# Enhancing oscillations in intracranial electrophysiological recordings with data-driven spatial filters

**Natalie Schaworonkow**[1]*, **Bradley Voytek**[1,2,3,4]

**1** Department of Cognitive Science, University of California, San Diego, California, United States of America,
**2** Halıcıoğlu Data Science Institute, University of California, San Diego, California, United States of America,
**3** Neurosciences Graduate Program, University of California, San Diego, California, United States of America,
**4** Kavli Institute for Brain and Mind, University of California, San Diego, California, United States of America

* nschaworonkow@ucsd.edu

## Abstract

In invasive electrophysiological recordings, a variety of neural oscillations can be detected across the cortex, with overlap in space and time. This overlap complicates measurement of neural oscillations using standard referencing schemes, like common average or bipolar referencing. Here, we illustrate the effects of spatial mixing on measuring neural oscillations in invasive electrophysiological recordings and demonstrate the benefits of using data-driven referencing schemes in order to improve measurement of neural oscillations. We discuss referencing as the application of a spatial filter. Spatio-spectral decomposition is used to estimate data-driven spatial filters, a computationally fast method which specifically enhances signal-to-noise ratio for oscillations in a frequency band of interest. We show that application of these data-driven spatial filters has benefits for data exploration, investigation of temporal dynamics and assessment of peak frequencies of neural oscillations. We demonstrate multiple use cases, exploring between-participant variability in presence of oscillations, spatial spread and waveform shape of different rhythms as well as narrowband noise removal with the aid of spatial filters. We find high between-participant variability in the presence of neural oscillations, a large variation in spatial spread of individual rhythms and many non-sinusoidal rhythms across the cortex. Improved measurement of cortical rhythms will yield better conditions for establishing links between cortical activity and behavior, as well as bridging scales between the invasive intracranial measurements and noninvasive macro-scale scalp measurements.

**Data Availability Statement:** All electrophysiological recording files are available from the "Library of human electrocorticographic data and analyses" database (https://purl.stanford.edu/zk881ps0522).

## Author summary

Invasive electrophysiological recordings of human brain activity offer the unique ability to measure multiple, simultaneously active brain rhythms. Analyzing brain rhythms is complex due to the fact that different oscillations often overlap in space and time. Here we explore human resting state invasive electrophysiological recordings by using spatial filters, which combine information from all available recording electrodes to specifically

**Funding:** This work was supported by the Whitehall Foundation (http://www.whitehall.org/recipients/) grant 2017-12-73 awarded to BV, the NIH National Institute of General Medical Sciences (https://www.nigms.nih.gov) grant R01GM134363-01 awarded to BV, and a UC San Diego Halıcıoğlu Data Science Institute (https://datascience.ucsd.edu) Fellowship awarded to BV. The funders had no role in study design, data collection and analysis, decision to publish, or preparation of the manuscript.

**Competing interests:** The authors have declared that no competing interests exist.

extract oscillations with high signal to noise ratio. Using this technique, we explore variability in oscillation presence across subjects, the spatial spread and waveform shape of oscillations. We find that participants differ a lot in presence of oscillations, even when the recording electrodes have similar placement. We find that oscillations exhibit spatial spread exceeding the distance between electrodes and that the waveform shape of oscillations in different brain regions can be highly deviating from a sine wave.

This is a *PLOS Computational Biology* Methods paper.

## Introduction

Invasive, intracranial electroencephalography (iEEG) recordings from patients undergoing epilepsy monitoring have been tremendously valuable for examining neuronal activity. This is because iEEG provides both high temporal and spatial resolution that is impossible to achieve using solely noninvasive human neuroimaging [1, 2]. There are different types of recording electrodes for iEEG, using electrodes arranged in grids that are commonly referred to as electrocorticography (ECoG) or using electrodes arranged along a linear array, which is referred to as stereoencephalography (sEEG). Because of the superior spatial and temporal resolution of iEEG, combined with the possibility of simultaneous single-neuron recordings from humans [3], these rare recordings provide a bridge between human cognition and decades of animal electrophysiology. The recordings display myriad types of complex activity, for instance prominent rhythms [4] in several frequency bands, overlapping in time and space. Cortical rhythms have been examined during resting-state activity [5, 6] as well as during tasks [7–10]. The rhythms show distinct spectral peaks, for instance in alpha- and beta-frequency range, distinct spatial distribution across rhythm types, for instance with beta-bursts prominent in the precentral gyrus, and the sensorimotor mu-rhythm in the postcentral gyrus [11]. Theta-rhythms are visible to a greater extent in invasive recordings, whereas in non-invasive recordings theta is mostly limited to mid frontal areas [12]. Rhythms show distinct task-related modulation and intricate waveforms strongly deviating from sinusoids, with these non-sinusoidalities potentially providing improved physiological interpretability beyond oscillation power alone [13].

Because the coexistence of different types of neural activity leads to superposition on the signal recorded with electrodes, many different methodological approaches exist to untangle distinct activity sources from electrode signals. One can leverage the multivariate structure of iEEG recordings, in which a number of electrodes are placed on the cortical surface to acquire time series data, toward this end. Each electrode picks up a mixture of signals from different types of cortical sources, determined by location and orientation of the generating sources and the biophysical properties of the tissue. Fig 1 illustrates the underlying data model for iEEG. This spatial mixing is given by the forward model and is assumed to be linear here [14].

For noninvasive electrophysiological recording techniques such as electroencephalography (EEG) and magnetoencephalography (MEG), source reconstruction techniques are commonly used to extract independent activity sources from sensor space data [15]. Many approaches can be framed as the estimation of spatial filters that satisfy pre-defined optimization criteria, taking into account either biophysical constraints given by cortex morphology or statistical properties of the signals, which for instance are considered when computing principal or independent component analysis. A spatial filter allows computation of a new, filtered signal trace

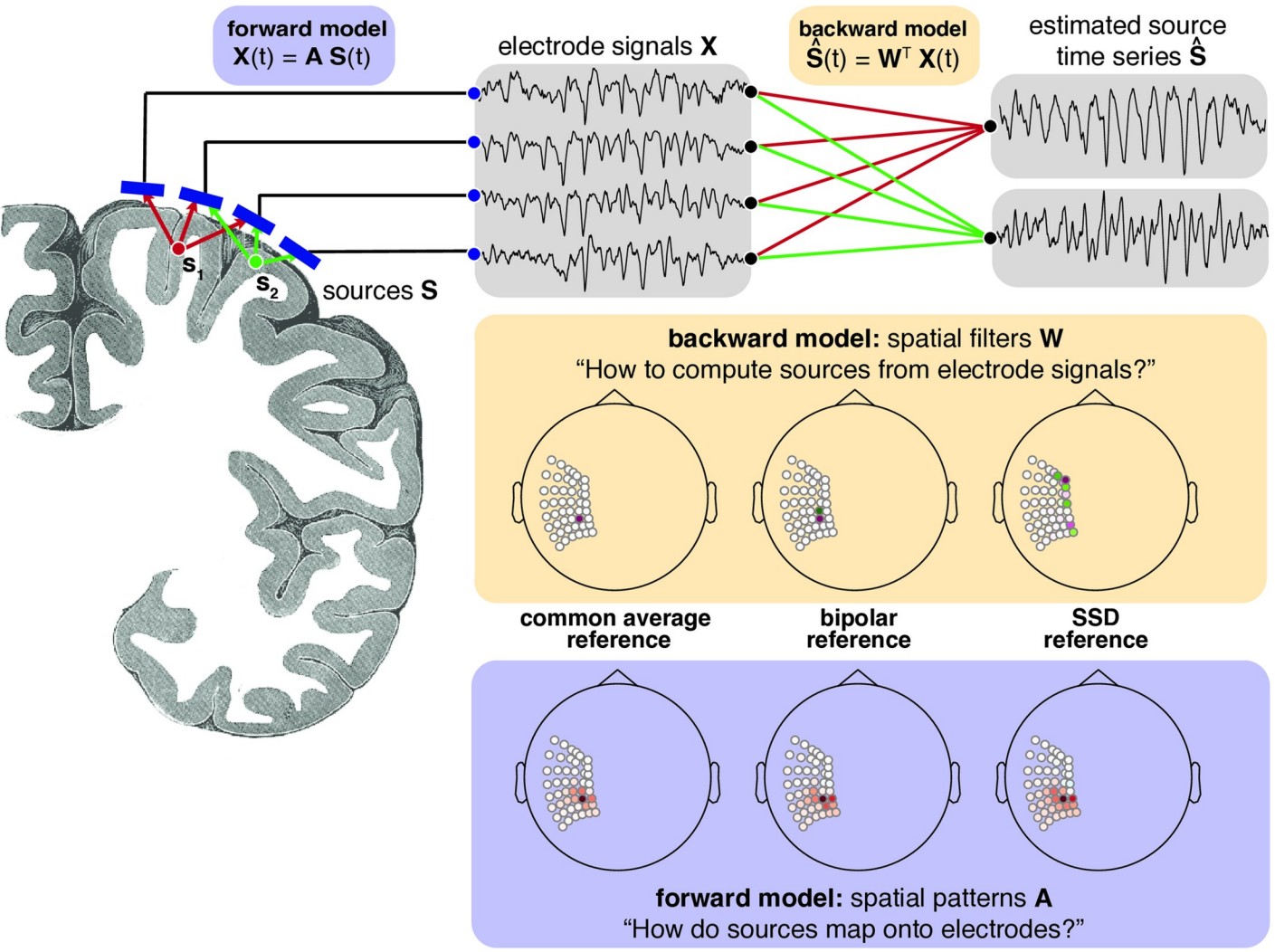

**Fig 1. Generative linear model of electrophysiological data.** Sources **S** in the gray matter mix according to the forward model **A** with the corresponding propagation of currents through tissue to the electrodes. The resulting signals **X** are recorded with electrodes placed on the cortical surface. The objective is to estimate the source time series $\hat{\mathbf{S}}$ from the electrode signals **X** with a backward model **W**. Three different backward models are illustrated with one specific example of their respective corresponding spatial filters and patterns. While the spatial filters can look quite different from each other, the spatial patterns point to a similar spatial origin of the extracted signal. Image source for coronal cut: public domain Gray's anatomy plate 718 [28].

using a weighted sum of all other electrodes. For each time point, the dot product of the electrode data with the spatial filter is taken to yield the corresponding entry for the source trace. The spatial filter vector is the same for all time points and this operation can be performed efficiently by matrix multiplication.

For iEEG recordings, source reconstruction has mostly been employed in the context of localizing epileptic seizure focus, both with biophysical modeling [16–19] and approaches using independent component analysis [17, 20–23]. But in contrast to non-invasive electrophysiological methods, data referencing techniques dominate for iEEG.

Data referencing can be viewed as the application of a particularly simple spatial filter. For instance, in the case of a bipolar filter, the spatial filter is a vector with as many entries as electrodes, containing weights -1 and +1 for two selected electrodes and zero for all other electrodes. The two most prevalent methods for referencing iEEG data are to apply either a

common average reference, with the aim to minimize common noise or distal activity, or to use bipolar reference, with the aim to extract locally generated signals. While a referencing approach is computationally simpler than an approach involving biophysically or statistically constrained source reconstruction, the referencing choice will highly impact the dynamics present in the resulting signal [24–27].

For examining high-frequency activity, an electrode-based approach (using a standard common average or bipolar reference) seems to be justified because of limited spatial spread of high-frequency signal content not exceeding inter-electrode distance [29, 30], with sub-centimeter functional specificity [31]. In contrast to that, activity in lower frequency ranges displays an increased spatial spread, showing a high degree of correlation between neighboring electrode locations depending on oscillation frequency [10, 32]. Because of the spatial spread, it is expected that different rhythms contribute to activity of several electrodes due to spatial superposition. Therefore, multivariate separation techniques may improve measurement of cortical rhythms also in iEEG, as for instance was examined using independent component analysis in [33].

Here, we explore a data-driven spatial filtering method, spatio-spectral decomposition (SSD) for specifically extracting oscillatory sources in iEEG data. This technique, based on generalized eigenvalue decomposition, has been shown to be superior to independent component analysis in EEG for extraction of oscillatory sources [34]. The SSD approach estimates distinct putative neuronal sources from the summation activity recorded via the electrodes, i.e., it estimates a backward model in the form of spatial filters with the optimization constraint focussed on a specific frequency band of interest, in order to best measure the temporal dynamics of oscillations in that band and their associated features of interest The focus of this approach here is primarily on estimating the source time series. The estimated source time series are subsequently referred to as components. Information about the location of a source is only indirectly provided through the computation of spatial patterns.

It is important to make a distinction between spatial filters and the spatial patterns associated with each filter. A spatial filter assigns a weight to each electrode that quantifies how much each electrode contributes to the calculation of an extracted component. A spatial filter is generally not interpretable [35], in the sense that the magnitude of the weights directly reflects the contribution of the source to the spatially filtered signal, as a large spatial filter weight may also be related to cancellation of noise, for instance.

Once the spatial filter weights are calculated, one can examine the spatial structure of each component by computing the spatial patterns, with each pattern reflecting the mapping of sources onto measured electrode signals. This quantifies the strength and polarity of a putative source signal on all electrodes. For instance, in Fig 1, for the bipolar referencing only two electrodes contribute to the calculation of the component. But due to the fact that neighboring electrodes exhibit signal correlation to the involved electrodes due to spatial spread, information about this source is also present in the vicinity of the two electrodes used for calculation of the bipolar derivation. Therefore, the associated spatial pattern has intermediate coefficients around the involved electrodes. The spatial patterns can for instance be computed by matrix inversion of the spatial filters. It can be seen that although the spatial filters in Fig 1 have different structure respectively, the associated spatial patterns are quite similar, reflecting a source originating in the sensorimotor region.

In this article, we use spatial filters to investigate rhythms present in mainly the alpha and beta-frequency bands in human iEEG recordings. We illustrate two aspects of measuring oscillations in iEEG data. First, that the activity spread of individual rhythms can exceed inter-electrode distance, with single rhythms contributing to several electrodes. Second, that spatial mixing of rhythms in intracranial recordings can affect the oscillatory power of a given rhythm

as detected on the electrodes and alter its non-sinusoidal waveform shape. We demonstrate how spatial filtering can identify rhythms that otherwise may not be apparent in the data due to masking by other stronger oscillatory contributions, from low signal-to-noise ratio (SNR), and/or from destructive interference. We also extract dominant rhythms in a resting state dataset with spatial filters and discuss variability across participants in presence of detected rhythms. Additionally, we illustrate how spatial filtering can be used as a powerful way to remove band-limited noise, without artefacts from temporal bandstop-filtering. While the employed spatial filtering methods are already used in analysis of noninvasive recordings, the aim here is to also highlight the specific benefits of using data-driven spatial filters for invasive electrophysiological recordings. Improved measurement of rhythms will aid bridging the scales from recordings obtained invasively to noninvasive recording techniques.

## Materials and methods

### Experimental recordings

We analyzed openly available datasets from a library of intracranial recordings [9]. We primarily used the dataset **fixation_pwrlaw** where participants fixated on a target location for several minutes, as our focus here is physiological rhythms in the resting state. For the single participant spatial mixing illustration in Fig 2 as well as S1 Fig and to show that spatial filtering preserves oscillatory task dynamics in S2 Fig we used the dataset **motor_basic**. In addition, we used one recording from the **faces_basic**-dataset to demonstrate the application of spatial filters for strip electrode recordings for several leads that are in close vicinity and one recording for removing noise with a specific spectral profile. We include the required ethics statement for each of those datasets in the following, as mandated by the data usage requirements of the data library.

**Ethics statements.** Data set **fixation_pwrlaw**: Ethics statement: All patients participated in a purely voluntary manner, after providing informed written consent, under experimental protocols approved by the Institutional Review Board of the University of Washington (#12193). All patient data was anonymized according to IRB protocol, in accordance with HIPAA mandate. These data originally appeared in the manuscript "Power-Law Scaling in the Brain Surface Electric Potential" published in PLoS Computational Biology in 2009 [36].

Dataset **motor_basic**: Participants in this dataset performed hand or tongue movements with timing based on a cue, with movement contralateral to placement of the recording grid. Cues were presented as written words in a 10 x 10 cm presentation window, within a distance of 0.75–1 m from participants. The analyzed dataset features a 39 year old female participant with a 5 x 5 electrode array, with an inter-electrode spacing of 10 mm, with a 4 mm diameter of each electrode. The sampling frequency was 1000 Hz, acquired with the sample recording system as above and hardware band-pass filtered in the same range.

Data set **faces_basic**: Ethics statement: All patients participated in a purely voluntary manner, after providing informed written consent, under experimental protocols approved by the Institutional Review Board of the University of Washington (#12193). All patient data was anonymized according to IRB protocol, in accordance with HIPAA mandate. These data originally appeared in the manuscript "Spontaneous Decoding of the Timing and Content of Human Object Perception from Cortical Surface Recordings Reveals Complementary Information in the Event-Related Potential and Broadband Spectral Change" published in PLoS Computational Biology in 2016 [37].

**Participants.** For resting state group analyses, the data from 20 participants was used. The mean age was 31.1±9.5 (mean±standard deviation), 9 female, 7 male. For four participants, age and gender information was not available.

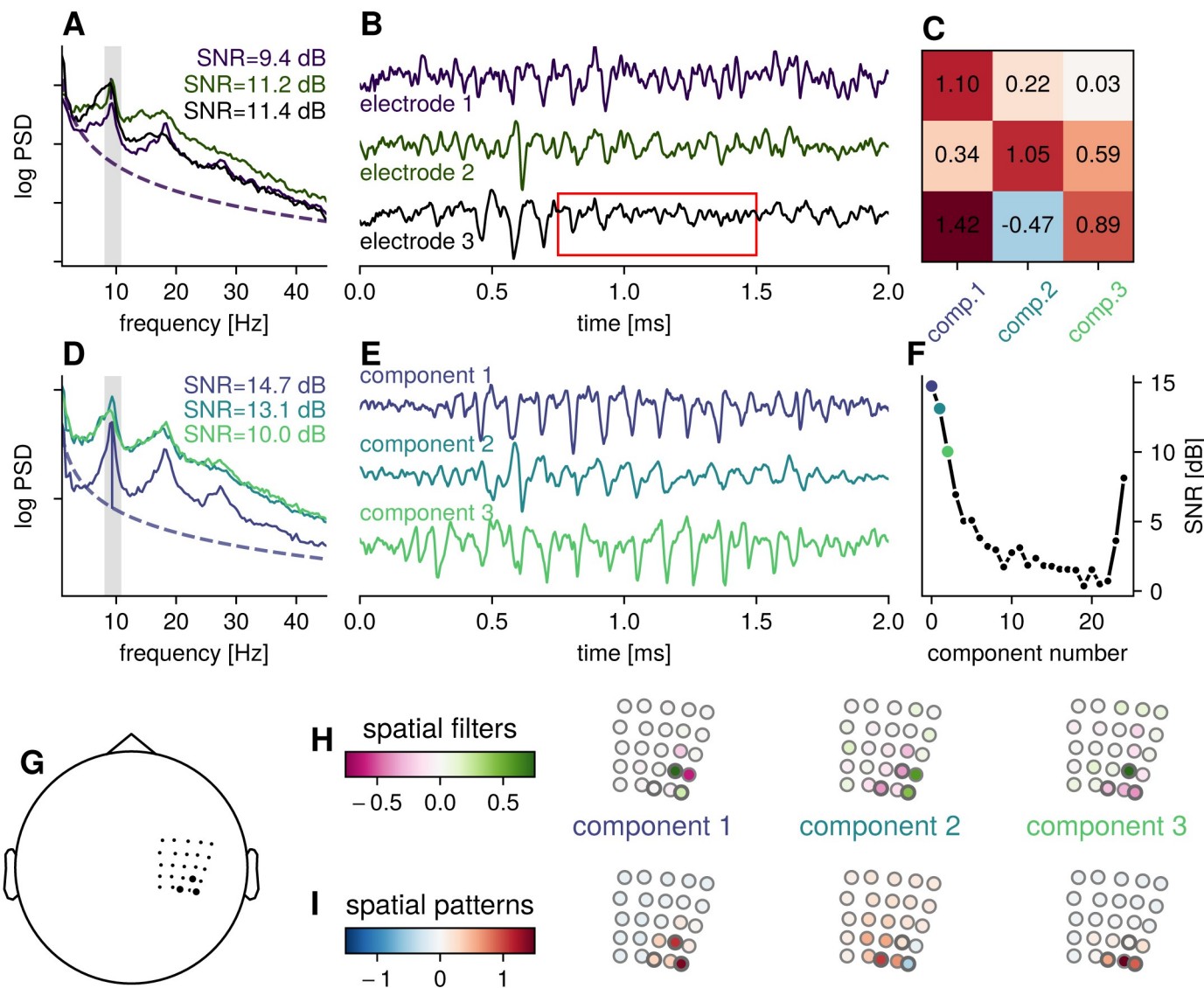

**Fig 2. Example: Spatial mixing of sensorimotor rhythms for one participant.** A) Power spectral densities for three channels along the sensorimotor strip. The gray bar indicates the frequency band defined as the signal contribution for estimating the SSD spatial filters. The power spectrum shows a peak frequency in the alpha-band, with additional harmonic peaks. The channels were selected a according to highest SNR in the chosen frequency range. B) The corresponding signal in the time domain showing oscillatory bursts in the alpha-band, amplitudes are normalized for comparison of time courses. The red box marks a time period in which less pronounced oscillations can be seen in the electrode signals, but the oscillatory power of the constituent SSD components is not decreased. C) Coefficients in the spatial patterns for the selected electrodes, i.e., electrode 2 can be approximated as a linear combination of: $e_2 = 0.34$ component$_1$ + 1.05 component$_2$ + 0.59 component$_3$. D) Power spectral densities for the first three components as estimated by SSD, showing a higher alpha-SNR, with less spectral peaks in flanking frequency bands. E) Time domain signal for the corresponding three components, showing pronounced sensorimotor bursts, normalized amplitudes for comparison of time courses. F) SNR per component, for all 25 components. The SNR drops off fast, only a number of components need to be inspected. For the components last in the sequence, the SNR increases as rhythms in flanking bands increase spectral power also in the band of interest. G) Approximate location of the ECoG-grid in head coordinates. The black markers highlight the electrodes shown in A) and B). H) Spatial filter coefficients showing similarity to bipolar and Laplacian-type filters. I) Spatial pattern coefficients showing focal contributions from sources along the sensorimotor strip.

**Experimental design and recording setup.** Dataset **fixation_pwrlaw**: The task was a fixation task where participants fixated on a fixation cross placed on the wall in three meters distance for several minutes (mean data length: 157±51 s). Intracranial recordings were made from subdural electrode arrays (mean number of electrodes: 60±12), with an inter-electrode spacing of 10 mm, with a 4 mm diameter of each electrode. For most participants, data was

available with a sample rate of 1000 Hz and was resampled to 1000 Hz for participants with a higher sampling rate. The recordings were made with Neuroscan Synamps[2] amplifiers (Compumedics-Neuroscan, San Antonio, TX) in conjunction with a clinical recording system (XLTEK or Nicolet-BMSI). A common ground and reference electrode, placed on the scalp, was used. A hardware band-pass filter from 0.15 Hz to 200 Hz was applied.

Dataset **motor_basic**: Participants in this dataset performed hand or tongue movements with timing based on a cue, with movement contralateral to placement of the recording grid. Cues were presented as written words in a 10 x 10 cm presentation window, within a distance of 0.75–1 m from participants. The analyzed dataset features a 39 year old female participant with a 5 x 5 electrode array, with an inter-electrode spacing of 10 mm, with a 4 mm diameter of each electrode. The sampling frequency was 1000 Hz, acquired with the sample recording system as above and hardware band-pass filtered in the same range.

Dataset **faces_basic**: Participants in this dataset performed a simple visual discrimination task. The electrodes had 4 mm diameter and 10 mm inter-electrode spacing, with silastic embedding. We selected three electrode leads over the left parietal hemisphere for analysis. The sampling frequency was 1000 Hz, acquired with the sample recording system as above and hardware band-pass filtered in the same range, the signals were measured with respect to a scalp reference and ground.

## Data analysis

Data analysis was performed using Python in conjunction with MNE v.0.20.4 [38]. Analysis code necessary to produce the figures in the manuscript from raw data is available at: https://github.com/nschawor/ieeg-spatial-filters-ssd.

**Spectral analysis and parametrization.** Power spectra were calculated with Welch's method (3 s window length, 0% overlap). The spectral parameterization method and toolbox of [39] (version 1.0.0) was employed for determination of peak frequencies. In this method, the power spectrum is modeled as a superposition of aperiodic and oscillatory components, which allows to distinguish between oscillatory and aperiodic contributions to the power spectrum. The power spectrum $P(f)$ for each frequency $f$ is expressed as:

$$P(f) = L(f) + \sum_n G_n(f).$$

(1)

With the aperiodic contribution $L(f)$ expressed as:

$$L(f) = b - \log[f^\chi],$$

(2)

with a constant offset $b$ and the aperiodic exponent $\chi$. When the power spectrum is plotted on a log-log axis, the aperiodic exponent $\chi$ corresponds to the slope of a line. Each oscillatory contribution $G_n(f)$ is modelled as a Gaussian peak:

$$G_n(f) = a_n \exp\left[-\frac{(f - \mu_n)^2}{2\sigma_n^2}\right],$$

(3)

with $a_n$ as the amplitude, $\mu_n$ as the peak frequency and $\sigma_n$ as the bandwidth of each component. The number of oscillatory components is determined from the data, with the option to set a maximum number of components as a parameter. The model assumption is that oscillatory and aperiodic processes are separable. Settings for the spectral parameterization algorithm were: peak width limits: (0.5, 12.0); maximum number of peaks: 5; minimum peak amplitude exceeding the aperiodic fit: 0.0; peak threshold: 2.0; and aperiodic mode: 'fixed'. Here, we only

extracted the peak frequencies and bandwidths for each electrode of each participant, discarding the aperiodic exponent.

**Calculation of spatial filters.** We estimate spatial filters via spatio-spectral decomposition (SSD) [34], which specifically maximizes spectral power in a frequency band of interest (for example from 8–12 Hz), while minimizing spectral power in flanking frequency bands (for example from 6–7 Hz as well as 13–14 Hz). This procedure enhances the height of spectral peaks over the 1/f-contribution, exploiting specifically the typical narrowband peak structure of neural oscillations. The underlying data model for the method assumes that the measured time series $\mathbf{X}$ (a matrix with $t$ samples and $k$ electrodes) constitute a linear superposition of signal $\mathbf{X_S}$ and noise $\mathbf{X_N}$ contributions in the data. In the particular case, signal here means oscillations in a narrow frequency band, while noise represents the signal in the neighboring frequency bands. The estimation procedure uses temporally band-pass filtered activity, centered on a peak frequency with a specified bandwidth. The choice of peak frequency and bandwidth was informed by spectral parametrization of signals from all electrodes. Following the original paper [34], we use a $4^{\text{th}}$ order Butterworth filter for temporally filtering the respective signal and noise contributions. The covariance matrices across electrodes of the signal and noise contributions are calculated on the basis of the band-pass filtered electrode activity.

$$\mathbf{X} = \mathbf{X}_S + \mathbf{X}_N$$
$$\text{signal covariance } \mathbf{C}_S = \mathbf{X}_S^T \mathbf{X}_S \text{ with } \mathbf{C}_S \in \mathbb{R}^{k \times k}$$
$$\text{noise covariance } \mathbf{C}_N = \mathbf{X}_N^T \mathbf{X}_N \text{ with } \mathbf{C}_N \in \mathbb{R}^{k \times k}$$

The objective is to find a spatial filter $\mathbf{w}$, which maximizes the power of the projected signal $P_S$, while minimizing the power of the projected noise $P_N$.

$$\text{SNR}(\mathbf{w}) = \frac{P_S}{P_N} = \frac{\text{var}(\mathbf{w}^T \mathbf{X}_S)}{\text{var}(\mathbf{w}^T \mathbf{X}_N)} = \frac{\mathbf{w}^T \mathbf{C}_S \mathbf{w}}{\mathbf{w}^T \mathbf{C}_N \mathbf{w}}$$

This Rayleigh quotient can be transformed into a generalized eigenvalue problem, which allows efficient and fast computation. In matrix form, the above equation can be written as:

$$\mathbf{C}_S \mathbf{W} = \mathbf{C}_N \mathbf{W} \mathbf{\Lambda}$$

where $\mathbf{W}$ is the matrix of all spatial filters with individual filters as columns, and $\mathbf{\Lambda}$ is the unity matrix with the corresponding eigenvalues on the diagonal. While the spatial filters are estimated with the aid of covariance matrices obtained from narrowband activity (from the narrowband activity defined as signal as well as the flanking narrowband noise), the spatial filters are then applied on the broadband activity recorded by the electrodes $\mathbf{X}$ to yield the component time series $\hat{\mathbf{S}} = \mathbf{W}^T \mathbf{X}$. The data can be reconstructed using $\mathbf{X} = \mathbf{W}^{-1} \hat{\mathbf{S}} = \mathbf{A} \hat{\mathbf{S}}$, where the inverse of the spatial filter matrix $\mathbf{W}^{-1}$ constitutes the matrix of spatial patterns $\mathbf{A}$.

Applying the spatial filters to the broadband activity ensures that features of activity originating from the same spatial location will also be extracted by the spatial filter, for instance the harmonics of a non-sinusoidal signal. The number of components returned by SSD is equal to the number of electrodes, with the components ordered by relative SNR in the frequency band of interest. In contrast to PCA, the first few SSD components only capture a small fraction of global variance, as the method is focused on maximizing variance in a specific frequency band. PCA has strong constraints, and can only return a spatial filter matrix $\mathbf{W}$ which is orthogonal, i.e., for which $\mathbf{W}^T \mathbf{W} = \mathbf{Id}$ needs to be satisfied. Therefore $\mathbf{W}^T = \mathbf{W}^{-1} = \mathbf{A}$, which means that spatial patterns $\mathbf{A}$ are equal to spatial filters for PCA. As this results in spatial filters with high degree of smoothness between neighboring values, PCA will not be able to distinguish rhythms

in the same subspace. SSD and other generalized eigenvalue decomposition methods do not have this constraint. There, the following constraint needs to be satisfied $\mathbf{AW} = \mathbf{Id}$. Therefore, the spatial patterns are not generally equal to the spatial filters, as the inverse of a matrix is not generally equal to its transpose. This allows distinguishing sources in the same subspace, e.g. rhythms coming from the same cortical area in the same frequency band. In terms of amplitude of the individual components, they are independent of each other for PCA as well as SSD.

A technical note: certain preprocessing operations, like removal of ICA components, can result in a covariance matrix that does not have full rank. To determine whether the covariance matrices have full rank, the eigenvalue problem involving only the signal covariance $\mathbf{C}_S\mathbf{V} = \mathbf{V\Lambda}$ is solved. The rank $r$ is determined by calculating the number of eigenvalues that are not zero (above a small numerical threshold, $10^{-6}$). If the matrix is not full rank, SSD is computed on the expansion: $\tilde{\mathbf{C}}_S = (\mathbf{V}_{1:r}\mathbf{\Lambda})^T\mathbf{C}_S\mathbf{V}_{1:r}\mathbf{\Lambda}$, with $\mathbf{\Lambda}$ being the identity matrix, having $1/\sqrt{\lambda_i}$ on the diagonal, with eigenvalues $\lambda_i$ and $\mathbf{V}_{1:r}$ constituting the first $r$ eigenvectors. After solving the generalized eigenvalue problem, the resulting spatial filters $\tilde{\mathbf{W}}$ are multiplied with $\mathbf{V}_{1:r}\mathbf{\Lambda}$ to obtain the spatial filters in the original space: $\tilde{\mathbf{W}} = \mathbf{V}_{1:r}\mathbf{\Lambda}\tilde{\mathbf{W}}$.

The peak frequency of estimated SSD components can differ slightly from the target peak frequency used to define the signal contribution. Therefore, after spatial filter estimation, the peak frequency and the SNR of each component (spectral peak height exceeding the 1/f-contribution) was assessed by calculating the power spectra and parametrization of them with the same parameter settings as for the electrode signals. Components exceeding a SNR-value of 5 dB were retained. This will discard weak rhythms, but the main objective here is to identify rhythms using a common threshold in order to make comparisons across participants. This was also done to illustrate a caveat in iEEG analyses, which commonly involves pooling of electrodes across participants, i.e., underlying here is the assumption of similar SNR across participants. The value of the SNR threshold was chosen in accordance with our previous studies [40], which were set to examine temporally resolved features of oscillations, e.g., instantaneous phase.

**Calculation of spatial patterns.**   Spatial patterns $\mathbf{A}$ for interpretation of the spatial origin of the extracted component can also be obtained by multiplication of spatial filter matrix $\mathbf{W}$ with the covariance matrix calculated for the signal component in the frequency band of interest $\mathbf{C}_S$ [35]: $\mathbf{A} = \frac{1}{Z}\mathbf{W}^T\mathbf{C}_S$.

For appropriate scaling, the patterns are normalized by a scaling factor $Z = (\mathbf{W}^T\mathbf{C}_S)^+\mathbf{W}$, with $^+$ denoting the Moore-Penrose pseudoinverse, such that the product of spatial patterns and spatial filters will yield the identity matrix $\mathbf{A}^T\mathbf{W} = \mathbf{Id}$. This is required in order for the product of the patterns and source estimates $\hat{\mathbf{S}}$ to yield the electrode measurements $\mathbf{X}$.

To illustrate spatial spread of oscillatory components, we analysed the topography of spatial pattern coefficients. For each component, the absolute value of the associated spatial pattern coefficients was taken and the values were then divided by the maximum value. The maximum spatial pattern coefficient in a distance of 2.5 cm around the maximum (distance value determined by Euclidean distance) was extracted to assess contribution of a single component onto several electrode signals. We chose to limit the calculation to the immediate surrounding of the spatial maximum based on work from [32], who modelled the decrease in spatial correlation across electrodes using different function fits in high density ECoG data. As the spacing across electrodes in the dataset used here was too coarse to fit a function, we opted to quantify the decrease in spatial spread across space by the maximum spatial pattern coefficient in the vicinity of the spatial maximum.

**Waveform shape analysis.**   The bycycle toolbox [41] was used for detecting and quantifying burst features in the time domain, using the following steps: First, a narrow band-pass filter

(finite impulse response filter, peak frequency ±3 Hz) was used for identification of zero-crossings. With aid of zero-crossings, cycle features are determined on broadband filtered (1–45 Hz) data. All cycles that pass predefined criteria were classified as bursts. We used the following parameter settings for determining bursts, consistent across datasets: minimum of three present cycles, amplitude fraction threshold = 0.75, amplitude consistency threshold: 0.5, period consistency threshold: 0.5, monotonicity threshold: 0.5. An amplitude fraction threshold of 0.75 retains only the cycles exceeding an amplitude higher than the 75th percentile. The relatively high threshold was chosen to allow for improved measurement of asymmetries, as burst occurrence may be quite infrequent. Then, mean waveform features across burst cycles (e.g., voltage amplitude and cycle frequency) were calculated for each component. A main focus here was the measurement of waveform shape asymmetries, i.e., peak-trough asymmetry, where the fraction spent in peak time (time from rising flank zero-crossing to falling flank zero-crossing) differs from the fraction spent in trough time (time from falling flank zero-crossing to rising flank zero-crossing), as well as rise-decay asymmetry, where the time taken from peak to trough differs from the time taken from trough to peak.

**Noise removal with spatial filters.** For removing noise with a specific spectral profile, we estimate spatial filters for maximizing SNR around the frequency peak that should be removed, e.g., 60 Hz ± 1.75 Hz for line noise. For defining the contribution that should be minimized, i.e. the contribution that should remain in the cleaned data, we adjusted the used frequency ranges slightly: while previously only a narrow frequency range was used for defining the flanking frequency, here we adjusted the lower range of the flanking pass-band to be at 1 Hz, such that the activity across the whole frequency range should be considered to remain in the data. The adjustment of frequency borders is a benefit of SSD, as it allows for flexible incorporation of prior knowledge for estimation of spatial filters. After estimation of spatial filters, the components constituting line noise are subtracted from the raw signal with a linear operation:

$$\mathbf{X}_{\text{cleaned}} = \mathbf{X} - \sum_{j=1}^{N} \mathbf{a}_j \mathbf{s}_j$$

with $\mathbf{X}$ the raw signal matrix, $N$ the number of components to remove, $\mathbf{a}_j$ the spatial pattern associated with component $j$ and $\mathbf{s}_j$ the time course of the SSD component $j$. $N$ can be determined by inspection of the power spectra of the estimated components, and removing components iteratively until the noise level reaches a sufficiently low state.

## Results

### Several rhythms contribute to intracranial activity from single electrodes

First, we illustrate how activity taken from single intracranial electrodes shows a mixture of several different rhythms. Each electrode features sensorimotor bursts in the alpha-frequency range, as indicated by a peak around 10 Hz in the spectral domain (Fig 2A) and cycles with a period of approximately 100 ms in the time domain (Fig 2B). We compute data-driven spatial filters using narrowband activity in the alpha-frequency range defined as the signal contribution and flanking frequency bands defined as the noise contribution. The estimated spatial filters are then applied on broadband activity. The spectra and examples of the time domain activity of the three components with highest SNR are shown in Fig 2D and 2E. The components display an increase in relative SNR (peak amplitude height over 1/f-contribution), compared to the raw electrode signals. The ordering of the components is according to the SNR in alpha-range, with the strongest relative SNR rhythm shown first. The SNR for all components can be seen in Fig 2F showing that SNR is highest for the first component, and a fast drop off

in SNR. This ordering enables fast inspection, as only the first couple of components typically contain activity in the frequency band of interest with sufficient SNR.

Of note in the example is that the first component is of much smaller total power than the second and third components, as can be seen in the power spectrum. The large relative SNR of the oscillation stems in part from the low power of the 1/f-contribution. In electrode 1 and 3, this component has a large contribution, as evidenced by large coefficients of the spatial pattern. But due to contributions of other components with a higher overall amplitude, this component is obscured on the level of electrodes. Only the activity in the alpha frequency band and surrounding bands is used to estimate spatial filters, but since harmonics originate from the same spatial location, due to the non-sinusoidal nature of the oscillation, the application of the spatial filters retains the SNR in the harmonic bands when the spatial filter is applied on a broadband signal.

Examining the spatial patterns associated with each component (Fig 2I), it is evident that the multiple rhythms co-occur in a small area, which results in each component contributing to the activity of the electrodes as indicated by large coefficients in the spatial pattern. In the case of tangentially orientated sources, the sign of the contribution can switch between neighboring electrodes. Depending on the spatial mixing of those rhythms, they can cancel out or enhance each other during specific time periods, due to changes in their phase relationship. In time periods where components of comparable amplitude are phase-aligned, constructive interference takes place, resulting in a large amplitude of the electrode signal. In periods with a phase shift reaching $\pi$ or 180 degrees, with peaks of one component coinciding with troughs of another, destructive interference can result in a low electrode signal amplitude, even though oscillations are still present in the individual components (see time points marked with red box in Fig 2B). Therefore, by constructive and destructive interference, changes in the power of the electrode signal can reflect changes in synchronization of rhythms across space [42], and do not necessarily reflect changes in the oscillation strength of the source signals. Disentangling these different possible causes of changes in oscillatory power cannot be done based solely on activity that displays a large degree of spatial mixing and data-driven spatial filters may be helpful to distinguish such phenomena.

Examining the spatial filters for each component (Fig 2H), they resemble but also diverge from bipolar or Laplacian-type spatial filters. The spatial filter associated with component 1 has a bipolar-type form, but with the advantage that the direction along which the bipolar derivation is taken is learned from data. In S1 Fig we show the same traces for common average referenced electrodes, which shows higher SNR compared to the common reference, but still exhibits spatial mixing, as well as bipolar derivations in two directions (anterior to posterior and lateral to medial). Additionally, we show in S2 Fig that task-related temporal dynamics are preserved by SSD spatial filtering, in line with other referencing methods, showing high consistency. The main argument here is not that SSD will achieve the highest SNR, but that referencing to capture specific sources is dependent on properties of the source and that not all sources will be captured best by a fixed referencing scheme, whose utility may depend on the local cortex morphology. In that sense, using data-driven spatial filters uses information given by the multivariate structure of recordings to a greater extent.

## Improvement of signal to noise ratio for sEEG signals

There are many ways to reference stereoelectroencephalography (sEEG) recordings, ranging from monopolar, bipolar, common average or Laplacian referencing [25]. The choice of reference is a researcher degree of freedom. Fig 3 shows application of SSD for a recording consisting of three close-by sEEG leads, as shown in Fig 3A. Fig 3B shows time domain examples of

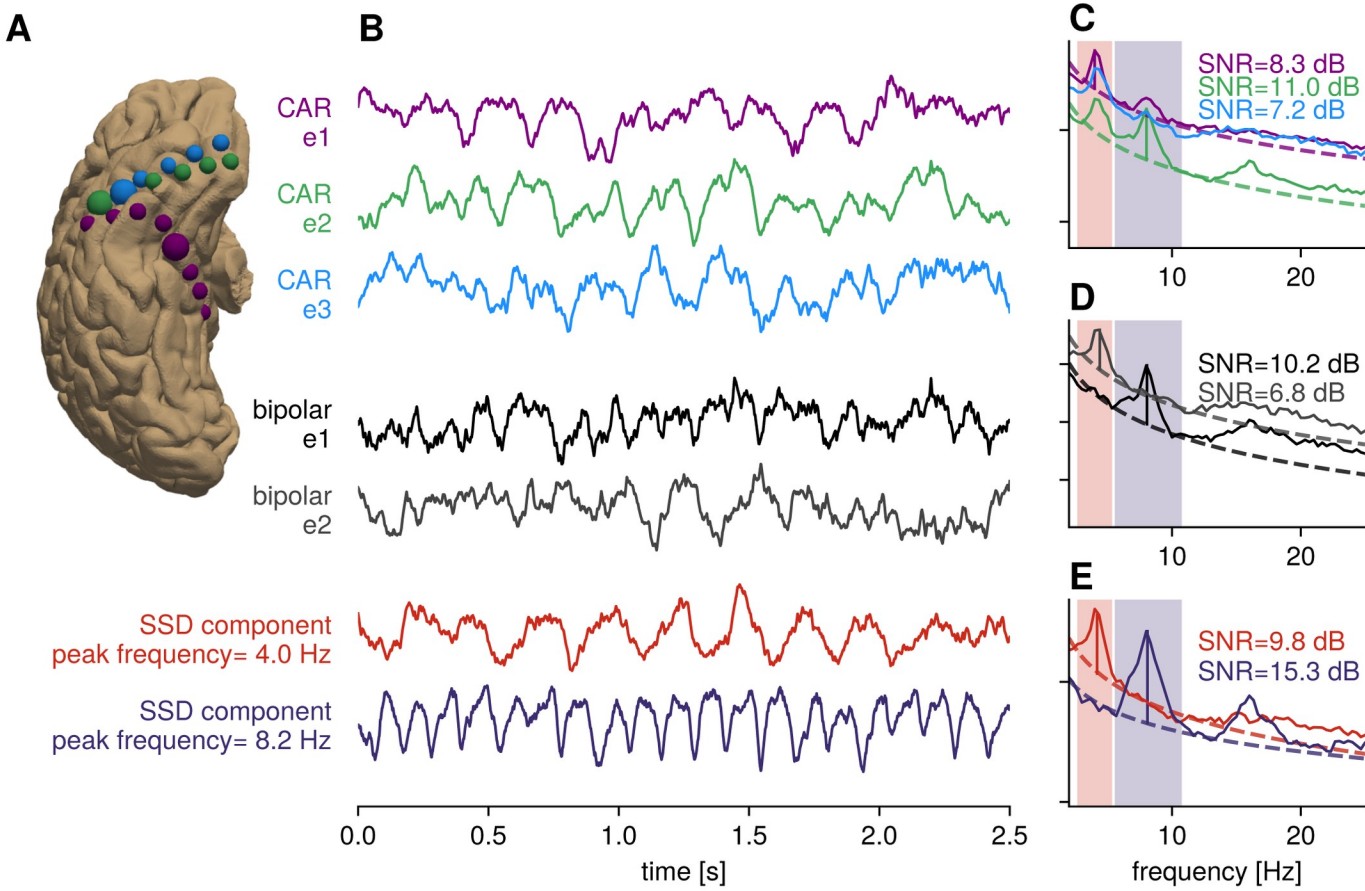

**Fig 3. Illustration: Increase of relative SNR for sEEG.** A) Three sEEG leads (blue, green, purple color respectively) plotted on cortical surface, ventral view. The electrodes highlighted with a larger circle size correspond to the colored traces. B) Time domain signals for common average referenced (CAR) signals, bipolar referenced signals and strongest SSD components for two selected peak frequencies, with bandwidth highlighted with colored boxes in spectral plots. C) Power spectral densities for common average referenced signals, showing multiple peak frequencies in the spectrum. D) Power spectral densities for bipolar-referenced signals. E) Power spectral densities for SSD components, showing an increased SNR over standard referencing.

electrode signals that were first common average referenced within a lead as well as examples for bipolar referencing (choosing a neighboring electrode on the same lead as the 2nd electrode), compared to the first SSD component for a peak frequency that was selected according to the spectrum of electrode signals, as is visible in Fig 3C for all three types of signals. It can be seen that SNR of rhythms of interest was improved in sEEG recordings by spatial filtering using SSD. In the case of the 8.2 Hz component, an enhancement of the spectral peak for the harmonic frequency is also visible, demonstrating the potential for SSD for isolating non-sinusoidal rhythms.

Additionally, all signals were submitted simultaneously to the SSD procedure, without subselection, making it possible to combine information from multiple leads and electrode configurations efficiently. While standard referencing techniques tend to enhance signals generated via a specific way, SSD is agnostic to the biophysical generation, and can be used more flexibly in this regard. For instance, monopolar and common-average referencing preserve correlation across channels to a higher degree than bipolar referencing [25]. This will influence measurement of rhythms, which can have a different spatial spread across subjects, depending on local cortex anatomy. So, while common average referencing highlights radial sources, bipolar referencing has a focus on locally generated activity and emphasizes bipolar sources. SSD will

extract components with maximum SNR agnostic about their spatial spread. In the next section, we further examine the spatial spread present empirically in iEEG data.

## Determining number and spatial extent of rhythms

Having demonstrated basic properties of spatial filtering with SSD, we turn to a number of physiological aspects to consider. Fig 4B shows time domain examples of two closely neighboring electrodes (black traces) with oscillatory activity in the same frequency band. The time domain activity of these electrodes looks similar, with prominent alpha-band oscillations. While these two electrodes look similar, it is unclear whether this is due to several independent rhythms with the same peak frequency, or one underlying source that is projecting onto both electrodes. Estimating spatial filters with SSD shows the existence of two

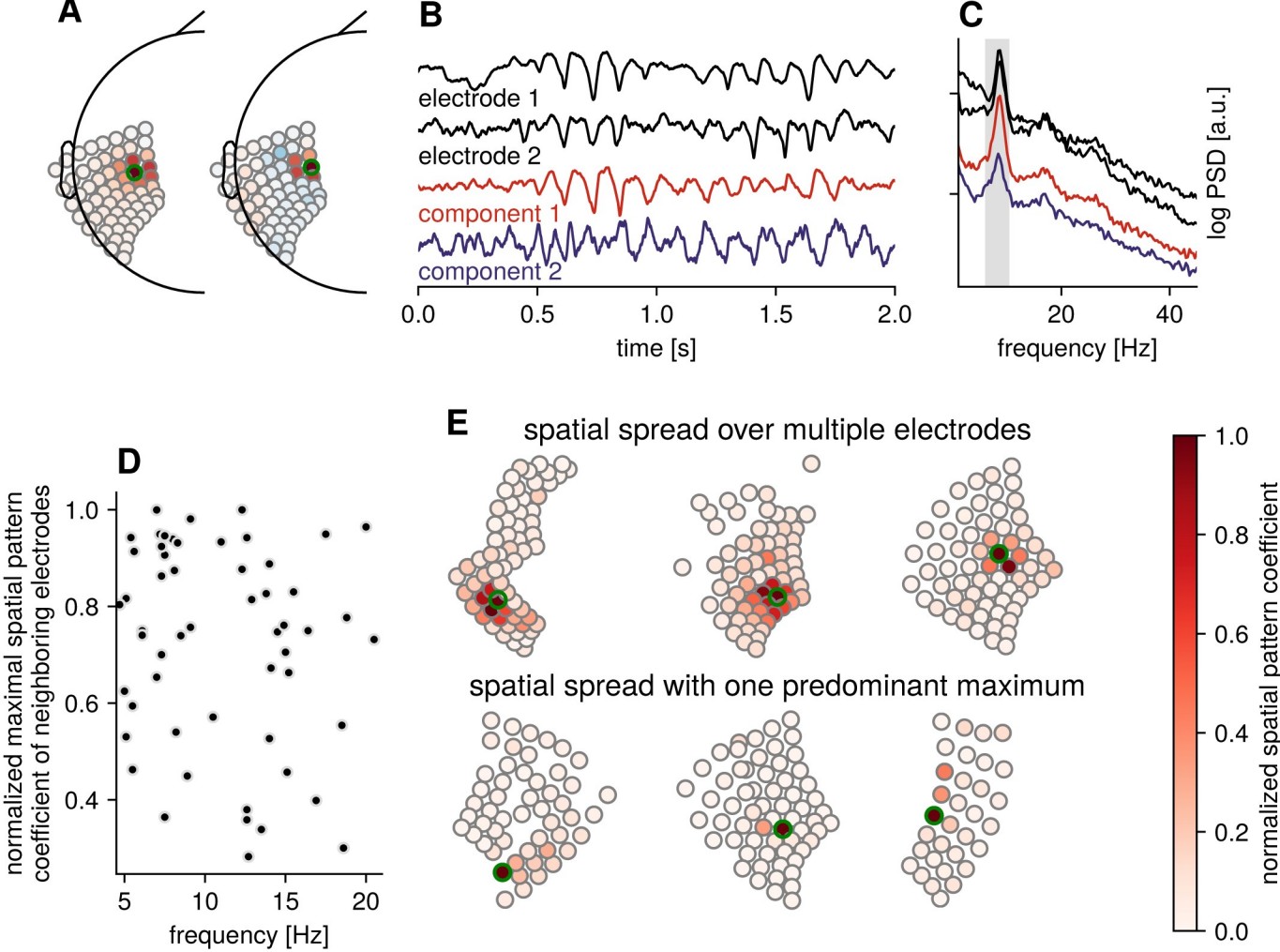

**Fig 4. Identifying independent sources.** A) Spatial patterns for two components with electrodes highlighted in green. B) Time domain activity for two neighboring electrodes (black) and the top SNR components for alpha range (gray span in the spectrum in C), showing that the oscillatory activity is largely captured by the first component, with a smaller alpha component in the second component that is otherwise masked in the electrode activity. C) Power spectral densities for electrode and component signals. D) Spatial spread for components with different peak frequency showing large variation. Each circle corresponds to one component. E) Example spatial pattern coefficients visualized on electrode grids, for high spatial spread (top row), where a component contributes to activity of many electrodes and low spatial spread (bottom row) with a single maximum. In contrast to A), the absolute value is plotted here to better illustrate the spread, regardless of polarity. The electrode with the largest coefficient is marked with a green circle.

components with alpha-rhythm activity, one strong alpha-rhythm (in terms of relative SNR) and a second alpha-rhythm with much smaller spectral power (Fig 4C), which would be masked in individual electrode traces. Of note is that the spatial spread of these rhythms exceeds inter-electrode spacing, with the same rhythm having large contributions to the activity of neighboring channels, as indicated by the spatial pattern coefficients showing high values for a cluster of electrodes. The components are closely overlapping in space as can be seen from the topography of spatial patterns (Fig 4A). By applying data-driven spatial filters, the rhythms close in frequency and space can be disentangled, enhancing the detectability of small amplitude rhythms, for instance.

To go beyond a single participant, we analyzed the coefficients of spatial patterns for our resting dataset. Fig 4D shows the spatial spread of different rhythms, as measured by the spatial pattern coefficients of electrodes neighboring the electrode with the maximum coefficient. It is visible that rhythms can have large contributions onto several electrodes, as indicated by neighboring channels with high coefficients. We show examples of rhythms with high spatial spread in Fig 4E, top row. A key point we want to highlight is that the spatial spread can also be small (see Fig 4E, bottom row), with only one singular maximum for one electrode relative to other electrodes. In such a case, a standard common average reference might be a sufficient but more simple approach for investigating rhythms. However, in the case of a rhythm of a large spatial spread across a large number of electrodes however, this rhythm may be attenuated when using a common average reference. Thus, the benefit of using data-driven spatial filters is that they may work in both cases, because the spatial correlations across electrodes for different present rhythms cannot be known a priori.

In general, signal decomposition techniques like SSD can be used for dimensionality reduction, keeping only the *N* components that contribute most strongly to the signal in the band of interest, and projecting out all other components to limit analyses to a specific subspace. The determination of which components to keep can be made using several different approaches, such as a threshold criterion based on 1/f-corrected SNR as in this article, a more local relative SNR-threshold criterion only focusing on the peak frequency band and neighboring flanking bands [34, 43], with the aid of a bootstrapping procedure [44], or based on physiological considerations such as focusing on rhythms originating from a specific location, which can be determined with aid of the spatial patterns. We want to stress that a criterion of the number of components to keep is dependent on the specific objectives of the study and needs to be carefully considered within the scope of those desired objectives. In the following, we employed a 1/f-corrected SNR criterion, as the aim was to quantify all dominant resting rhythms without any regional pre-selection.

## Variability of resting rhythms across the cortex

To demonstrate how data-driven spatial filters can be used for data exploration, we assess the resting rhythms in the frequency range of 5 to 20 Hz for different participants in an iEEG resting state dataset. For this analysis, spatial filters were computed separately for specific frequency bands, with the frequency ranges selected via spectral parameterization of electrode signal to identify putative oscillations that exhibit narrowband power above the 1/f-contribution. The components with SNR exceeding a specified threshold (>5 dB) were retained. Fig 5A shows the spatial distribution of different rhythms extracted with SSD for individual participants. The location of the respective component marker reflects the electrode location of the maximum spatial pattern coefficient. In general, there are several rhythms detectable in the 5 to 20 Hz range. But there is considerable variation in peak frequency and measured SNR

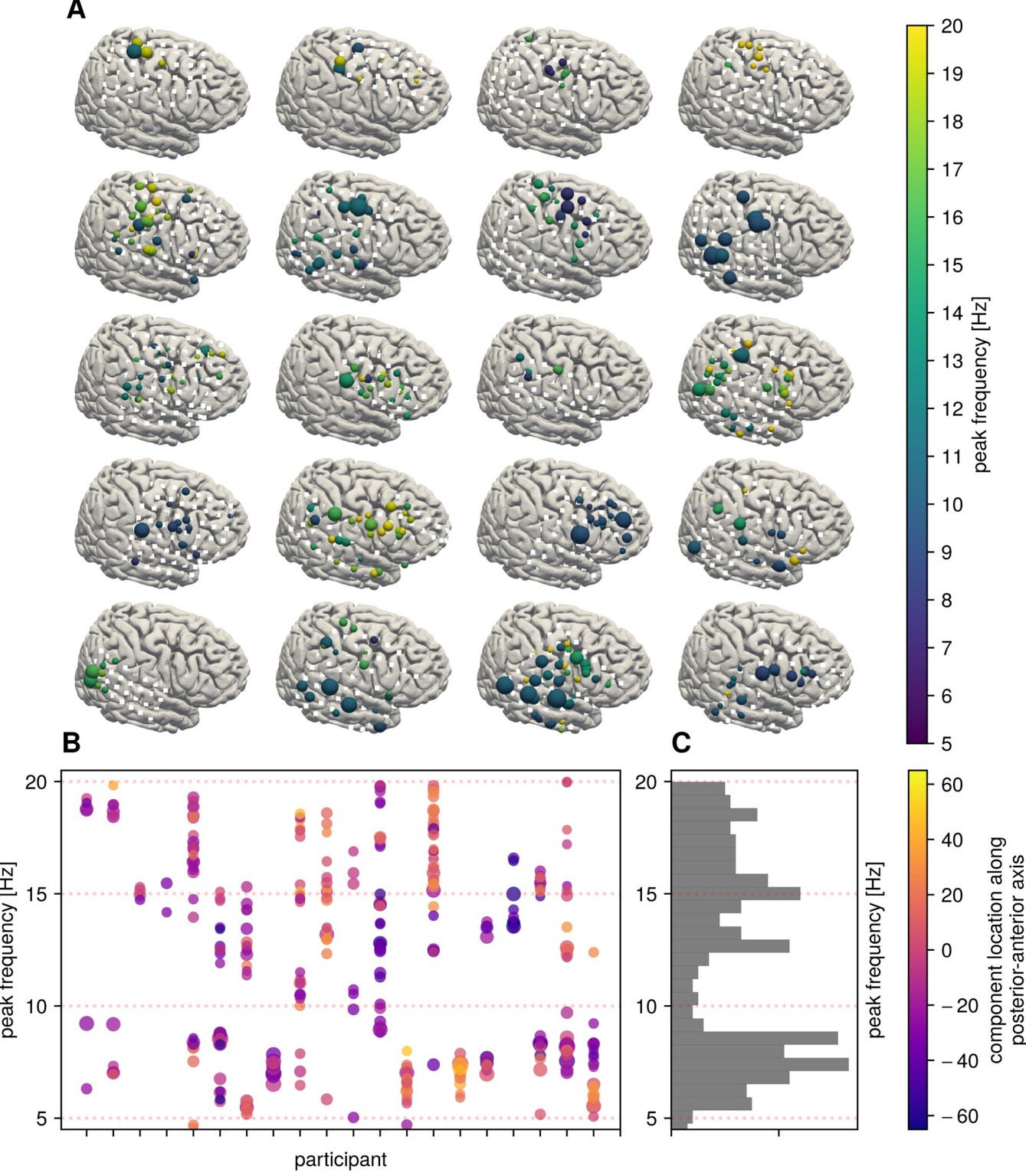

**Fig 5. Variability of resting rhythms across the cortex.** A) Each subplot shows the location of electrodes (white squares) on a template brain for one individual participant. Each sphere indicates an oscillatory component, with the size indicating 1/f-corrected SNR and the color indicating peak frequency of that component. If there is no sphere of a respective color in the vicinity of an electrode, no rhythm above the SNR-threshold could be detected in that frequency band. There is large variability between participants. For improved comparison across participants, all electrodes and rhythm locations were mapped onto the right hemisphere. Participants are ordered according to the mean z-coordinate across the electrode grid, to ease comparison. B) Each component is represented

as a circle, with y-position reflecting peak frequency and x-position reflecting participant ID. Color represents position along the posterior-anterior axis, with negative values reflecting most-posterior position. C) Histogram across all participants and components showing a relative lack of detectable 10 Hz rhythms.

across participants, as indicated by a variety of possible component arrangements across the cortex.

We also show the peak frequency of all identified components in Fig 5B and 5C. We find a distribution similar to [5], where there are more rhythms detected with a peak frequency around 7 as well as 16 Hz, and fewer rhythms with a peak frequency around 10 Hz, in contrast to non-invasive electrophysiological measurements. Note that because we use spectral parametrization on the spectra of SSD components, the peak frequency of SSD components can slightly vary from the peak frequency used as an input parameter for SSD. This distribution of peak frequencies is possibly related to the spatial bias of electrode placement, wherein most are placed over sensoriomotor and temporal areas, with less coverage over occipital areas, as determined by clinical needs. Additionally, the recording spans the duration of several minutes, during which time the participant's behavior is relatively unconstrained and variable. However, such a duration is typical for resting-state or baseline recordings across different studies. Another factor to consider, especially in iEEG data, is the fact that patients have the grids implanted for clinical reasons, with different pathologies and different medication status, which might contribute to the observed variability. Nevertheless, variability in peak frequencies and oscillatory SNR is also observed in non-invasive electrophysiological measurements. For iEEG, this might be more of a concern, since a smaller number of participants are usually included per study, compared to studies using non-invasive measurements. In the case of such small sample sizes, a single participant with a large amplitude, prominent rhythm across many electrodes may dominate the analysis due to the way that iEEG data are often pooled. This can result in a seemingly large effect in the group-average, despite only being present in a small number of participants.

The key point we aim to illustrate here is that rhythms are present with different SNR and variation in peak frequency across participants, and canonical oscillations of interest may or may not be detectable in individual participants in iEEG data, given the large between participant variability. The variability may contribute to inconsistent results, as temporal band-pass filtering of activity in a certain area within a predefined frequency band might not actually reflect oscillatory dynamics, but might capture only contributions from 1/f-components. Applying data-driven spatial filters can aid in verifying the presence and spatial origin of rhythms of interest, while also improving measurements of their precise temporal dynamics by increasing the SNR.

### Waveform shape and spatial mixing

Neural oscillations are often of a non-sinusoidal shape, for instance in the form of a pronounced arc-shape in the case of the sensorimotor mu-rhythm (e.g., see Fig 2, component 1). While waveform can be informative about neuronal processing [13], the detectability of waveform shape requires a high enough SNR to capture harmonic frequencies, which may not be detectable with a high level of 1/f-noise. While non-sinusoidality is also present in noninvasive signals, the difficulty there is that it is often obscured by spatial mixing and a low SNR. Therefore, invasive cortical recordings provide an excellent opportunity to study waveform shape. As the amplitude envelope of rhythms in the same frequency band tends to co-fluctuate [45] positively, such as when oscillations are present for one source, there are also oscillations present in neighboring sources, there is a risk of harmonics canceling out due to spatial mixing. For instance, a temporal shift of 12.5 ms constitutes a period of $\frac{1}{4}\pi$ for a 10 Hz alpha, but twice

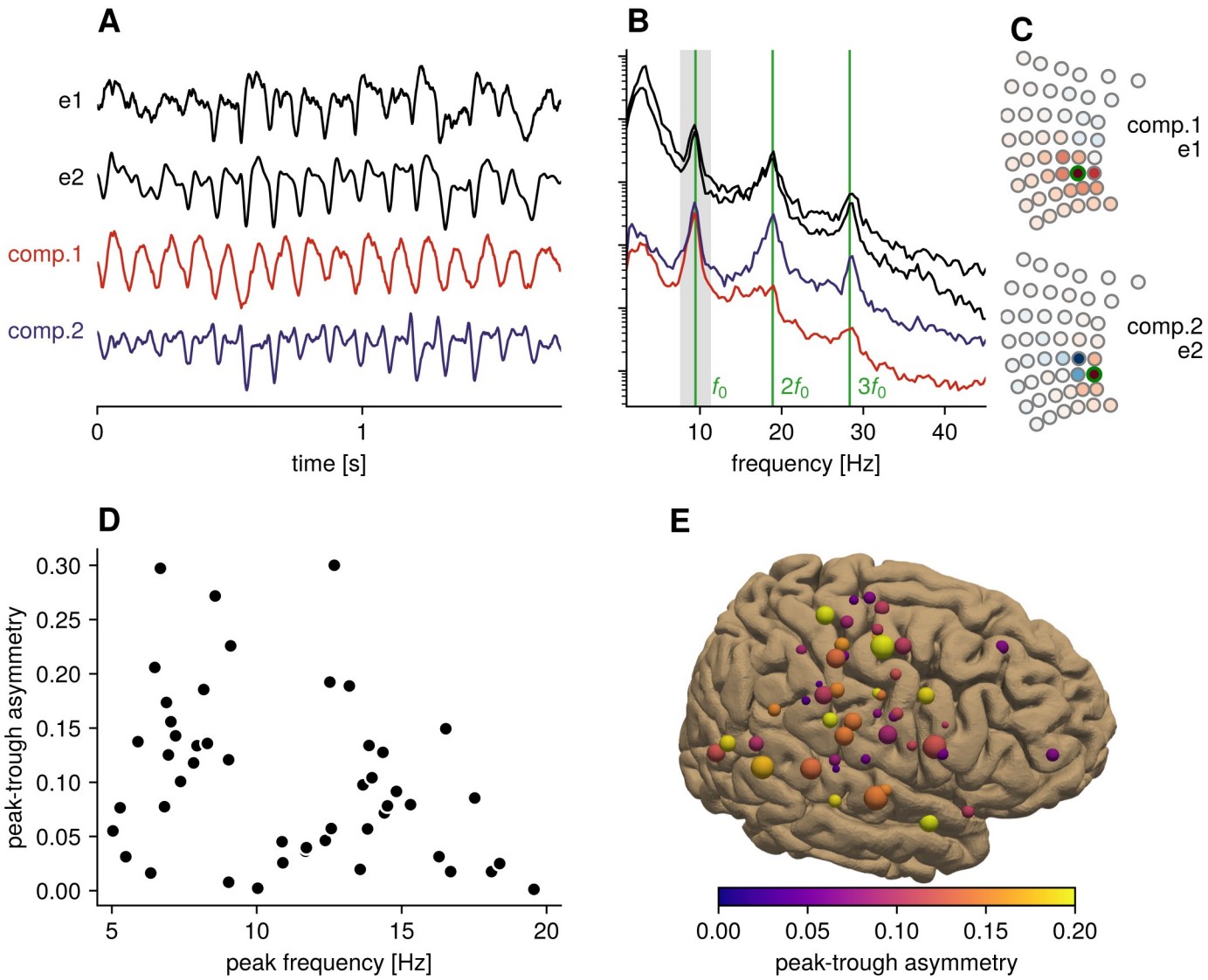

**Fig 6. Waveform shape of intracranial neuronal rhythms.** A) Neighboring rhythms with different waveform shape for two electrodes and two components estimated based on alpha band activity. B) Power spectral density for electrodes and components. The presence of harmonic spectral peaks at exact multiples of the alpha peak frequency indicates a non-sinusoidal waveform shape. The gray marked area corresponds to the frequency range defined as a signal for estimation of spatial filters. While both electrode signals show a peak in the beta-band, in component space the sharp beta-harmonic is largely captured by the second component, showing a spike-wave waveform shape, with the first component being a triangular waveform. C) Topographies for the first and second components showing a radial and tangential source distribution (respectively); the electrodes shown as traces in B are marked with green circles. D) Group-level assessment of waveform asymmetry, with intracranial recordings showing considerable peak-trough asymmetry in the waveform (where a peak-trough asymmetry value of 0 is indicating perfect symmetry). E) Peak-trough asymmetry values, plotted across the cortex, larger circles indicate larger SNR. Rhythms with high asymmetry can be found through-out the cortex.

the amount, $\frac{1}{2}\pi$, for a harmonic 20 Hz beta-rhythm, resulting in waveform changes solely induced by spatial smearing [42]. Therefore, using spatial filtering techniques can be beneficial to explore waveform properties in iEEG recordings.

Fig 6A shows an example where spatial demixing reveals two different rhythms with varying waveform shape properties. While the frequency spectra (Fig 6B) of the raw electrode traces activity traces look similar, waveform shape features are masked in the signals because different neighboring (Fig 6C) rhythmic components with different waveform shape

characteristics summate. The estimated components show a differentiation between a triangular rhythm and one that has a spike-wave shape (reminiscent of local field potential traces that can be found for instance in deep layers of macaques [46] or mice [47]). Differentiation between different waveforms may be difficult to make on the basis of signals highly impacted by spatial mixing.

Fig 6D illustrates that many ECoG rhythms in the 5–20 Hz frequency band have a non-sinusoidal waveform shape, which is information that could be taken into account to make inferences about underlying cellular physiology. Here, a peak-trough asymmetry value of 0.2 would mean a peak time of 60 ms and a trough time of 40 ms for an oscillation cycle of 100 ms length (a strong deviation from a 50/50 duty cycle). Non-sinusoidal rhythms can be present in various cortical locations, and in Fig 6E we show examples of high peak-trough asymmetry in the sensorimotor as well as temporal regions. Of note is that the triangular rhythm visible in Fig 6A is not asymmetric with respect to duty cycle, but still deviates from a sinusoid by showing sharpness around peaks and troughs. Therefore, construction of measures capturing waveform properties requires careful consideration regarding which aspects to measure, which may be different regarding physiological settings or disease pathologies of interest, e.g., peak-trough asymmetry for the sensorimotor mu-rhythm [48] or cycle sharpness in Parkinson's disease [49].

## Removing noise with a specific spectral profile using spatial filters

Another use case of spatial filters that we want to highlight here is in the removal of noise with a specific spectral profile in multichannel data. A prominent noise source in that respect is line noise with a high narrowband spectral peak at e.g., 50 or 60 Hz. Fig 7 illustrates the removal of noise from a raw ECoG recording that shows high levels of noise at 60 Hz across the majority of electrodes, as well as narrowband noise at 200 Hz that is of unknown origin (possibly caused by medical equipment). We estimate spatial filters to maximize SNR first for the 200 Hz noise, and subsequently project these components out from the raw signal by a linear operation. The signal after removal of noise is shown in Fig 7B. Fig 7C shows the signals after applying common average referencing; though the level of noise is attenuated, narrowband noise remains present in the signals. The benefit of using spatial filters to remove noise, as opposed to bandstop filtering, is that spatial filters do not cause distortions in the time domain around the noise frequency. This is especially important because signals in the frequency range of 50–70 Hz can be highly informative in ECoG, which makes preserving information in this frequency band of particular interest. Another benefit is the computational simplicity, the free parameters here are the peak frequency, the bandwidth around the peak frequency and the number of components to remove. The degree of attenuation can be selected by adjusting the number of noise components to remove, with a larger number yielding greater attenuation at the cost of potentially removing the signal of interest. For line noise removal with spatial filters, the ZapLine toolbox [50] provides several optimized routines. In ECoG, in addition to line noise, other noise sources can be present, and the flexibility of spatial filters allows for the efficient removal of noise with stationary spectral profiles.

The cost of this type of noise removal is the loss of dimensionality equal to the number of removed components, similar to the effect of applying a common average spatial filter. This loss of dimensionality is not of concern when a high number of electrodes are present, but would not be recommended for a small number of electrodes. While a common average spatial filter may work well if there is a common noise source that is manifesting in all electrode signals, using data-driven spatial filters allows for more flexibility if noise is not present in all signals. If too many noise-related components are removed, this bears the risk of removing signal

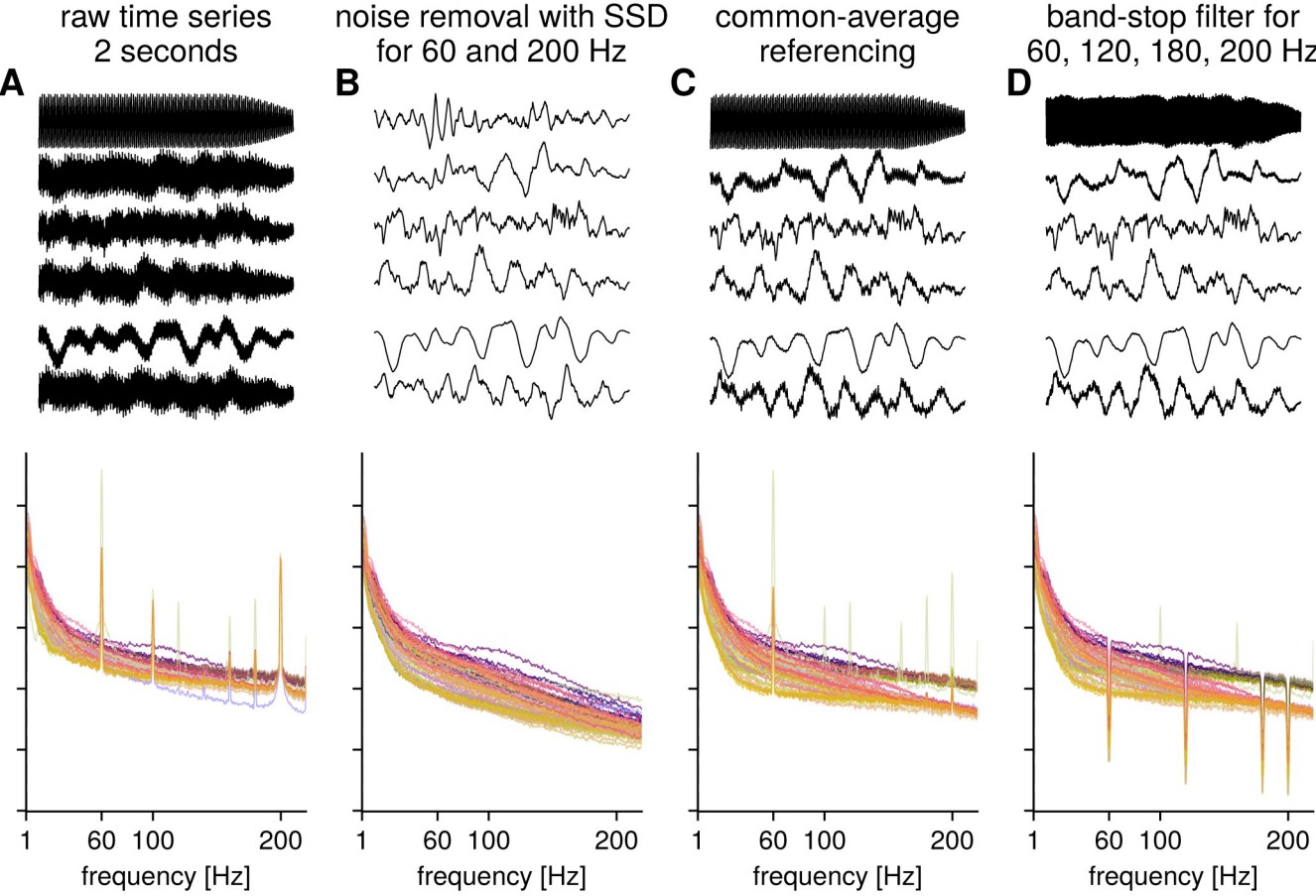

**Fig 7. Illustration: Removal of noise with spatial filters for ECoG data.** A) Time series of six electrodes and power spectra for raw ECoG recording for 87 electrodes, color code corresponds to electrode position, with neighboring electrodes having a similar color. B) Time series and power spectra after removal of components maximizing SNR for 60 Hz and 200 Hz spectral peaks. Note that there are no band-stop type artefacts in the spectrum since no temporal filtering was performed. C) Time series and power spectra after common average referencing. While the 200 Hz noise is largely attenuated, 60 Hz line noise still persists. D) Time series and power spectra after common average referencing and then band-stop filtering. The band-stop filters introduce artefacts in the spectral domain.

contribution that is of interest to the research question. It is therefore necessary to inspect the estimated noise components, for instance regarding the presence of spectral peaks in the frequency band of interest. If time-locked analyses are performed, noise components can also be inspected for the presence of time-locked contributions, to rule out reduction in valuable information by noise component subtraction.

## Discussion

In this article, we highlighted the benefits of using spatial filters for the extraction of neural oscillations in invasive electrophysiological recordings. Applying spatial filters that specifically optimize for oscillatory SNR in iEEG recordings, we assessed presence, spatial spread, variability and waveform shape of iEEG resting rhythms. SSD and other spatial filtering techniques can be a potential tool in the toolkit for researchers specifically interested in oscillations. As with all tools, careful consideration of the benefits and limitations has to be weighed against the increased complexity and freedom in parameter choices that might give way to potential false positives.

## General benefits of data-driven spatial filtering

Spatial filters can be used for distinct purposes in the study of neural oscillations: to identify rhythms in the frequency band of interest and improve their signal-to-noise ratio, to examine their correlational structure, as well as to denoise data without band-stop filters. On the continuum of using a common average reference spatial filter (potentially capturing mainly non-local activity) to local bipolar spatial filters (with a potentially non-optimal direction of the fixed derivation), the presented spatial filter technique represents a middle ground, extracting signals based on spatial spread as estimated from the data. This procedure results in reduced bias compared to a fixed reference choice.

The presented method belongs to a subclass of spatial filtering techniques that estimate a backwards model using solely the statistical properties of the signals recorded from the electrodes. The benefits of using statistical approaches like SSD, in contrast to biophysical modelling, is that no anatomical information or biophysical model is required for the estimation of the spatial filters, which strongly reduces the complexity of the procedure. While our demonstrations are mostly using ECoG data (as this was the predominant recording type present in the used dataset), the method can be similarly applied to sEEG data, as seen in Fig 3, for the benefit of combining information from different electrode leads. Whenever time series data from multiple electrodes is available, the method can be applied. The electrode locations are only needed for the interpretation of spatial patterns, but the source time series estimation is independent from the localization accuracy of the electrode positions.

The main benefit of SSD for the study of neural oscillations is that information about the signal structure at the frequency band of interest is incorporated, enhancing activity in that band of interest. This procedure is in contrast to independent component analysis or principal component analysis, which both maximize global objectives, which may not be optimal because components in specific frequency bands might only contribute a small amount to global variance. SSD results in an ordering of components according to SNR in the frequency band of interest, which reduces manual inspection and can facilitate data exploration. Additionally, SSD has few parameters and is computationally fast. The signal is defined using a temporal band-pass filter around the frequency of interest. The temporal filter requires a specification of frequency ranges for respective signal and noise contributions, for which the prior values can be derived from electrode power spectra. Even though only narrowband information is used for the estimation of spatial filters, the application on broadband data preserves information beyond these narrow frequency bands, such as waveform shape, as long as they originate from the same spatial location.

While we chose SSD as a spatial filtering technique for our illustrations, other types of generalized eigenvalue decomposition algorithms are available to solve specific objectives. For enhancing specifically oscillatory SNR, there are variants that maximize the spectral power in a frequency band of interest, compared to the total spectral power [51], with demonstrations for MEG/EEG as well as monkey ECoG and optical imaging given in [52] and which are benchmarked for EEG in [53]. Note that in the latter, the benchmark test for SSD only used the band-pass filtered activity for evaluation after application of spatial filters; in this way, many of the non-sinusoidal properties are lost. In our study, we compare the output of SSD using broadband filtered data to preserve nonsinusoidal waveform shape, and we expect other generalized eigenvalue decomposition methods aimed at amplifying oscillatory SNR to perform similarly to SSD when evaluated on broadband (versus narrowband filtered) data. The main aspect we want to highlight here is that generalized eigenvalue decomposition methods are highly flexible and permit interesting contrasts for maximizing/minimizing SNR along specific dimensions. For instance, in the case of task-based data, Common Spatial Patterns [54]

maximizes differences between conditions, for instance to investigate main contributions to task-related modulation. Source Power Correlation analysis [55] maximizes correlation with a target variable, for instance with reaction times. To consider extracting coupled neuronal sources, an extension of SSD termed Nonlinear Interaction Decomposition [56] has been suggested. If specific types of rhythms are of interest, SSD can also be used as a regularization technique before using other analysis methods to limit the analyzed signal to the frequency range of interest, for instance focusing on rhythms in the alpha band [43, 57].

## Benefits of data-driven spatial filtering for iEEG recordings

While the above listed benefits are general and also hold for applying data-driven spatial filtering for noninvasive electrophysiological signals, there are specific considerations when applying these methods for invasive electrophysiological recordings.

In contrast to noninvasive scalp recordings with standardized EEG electrode caps or sensor arrays in MEG, invasive electrodes are placed according to specific and heterogeneous clinical demands, and therefore do not conform to standardized positions. This complicates the incorporation of anatomical information into analyses. For biophysical source estimation/localization approaches, the accuracy will vary depending on electrode localization accuracy based on postimplantation CT imaging as well as accuracy of the estimated forward model. Reference-choice is also more variable in invasive recordings, with referencing often done to bone or a shank, as well as bipolar or monopolar referencing schemes, which may bias analyses. In the specific case of analyzing neural oscillations, using SSD may be beneficial, since it specifically utilizes information in the frequency band of interest, maximizing SNR in a more flexible way. In addition, nearby electrodes can have different SNR due to the placement of individual electrode contacts on cortical vasculature, which can influence the resulting signal in a frequency-dependent manner when using other common referencing schemes [58]. In addition, SSD allows for the use of information from several grids, or separate but close-by sEEG leads, which can convey improved SNR in contrast to using information from individual leads separately, as in the case of a fixed referencing scheme. The data-driven approach could also be helpful in integrating information from several types of electrodes, for instance in hybrid micro-macro electrode schemes [59].

While strong alpha-oscillations tend to dominate in non-invasive recordings, a more variable set of oscillations, with neighboring peak frequencies, is simultaneously detectable in iEEG recordings. To improve the detection of these typically smaller amplitude rhythms, spatial filtering might help to separate them from more dominant frequency rhythms, as SSD attenuates the signal contribution from flanking frequency bands, and therefore the separation of distinct sources in neighboring frequencies becomes possible with SSD.

## Physiological considerations: Spatial spread

Previous work has identified the degree of spatial correlations across invasively acquired field signals in varying frequency bands [29, 32, 60, 61], with estimates ranging from 400 $\mu m$ for the local field potential to several millimeters in the case of ECoG. Modeling work by [62] suggests that important consideration is the input correlation of the involved neuronal populations, where spatial spread grows as the degree of input correlations increases. While the spatial spread is limited for high-frequency signals, for low-frequency rhythms the spatial spread exceeds the interelectrode distance, which results in the same source contributing to several electrodes.

We showed examples of spatial spread as estimated from spatial patterns, ranging from small to large spread (Fig 4). The spread for a specific component and possible neighboring

signal sources cannot be known a priori and differs across sources and participants. Therefore spatial mixing will potentially obfuscate temporal dynamics of neighboring sources. Data-driven spatial filters can help to separate contributions of different sources onto corresponding electrodes, and spatial pattern coefficients can be used to visualize the composition of electrode activity as a linear combination of different sources. It would be of interest to evaluate whether higher-density ECoG grids with a smaller inter-electrode spacing than analyzed here would result in improved ability to separate rhythms in cortical areas where a large number of independent rhythms with different peak frequencies are present, such as along the sensorimotor strip.

### Physiological considerations: Waveform shape

The analysis of rhythms using only oscillatory amplitude and frequency discards a lot of potentially valuable physiological information. Using waveform shape measures can enable a more refined look on cellular generation mechanisms and functional relevance of rhythms [63, 64], however the detection of non-sinusoidal features of waveforms requires high SNR, making the analysis of non-sinusoidal waveforms difficult. Because of this, intracranial recordings are well suited for investigations of waveform shape. We show that rhythms as detected in ECoG can be highly non-sinusoidal in a variety of cortical areas. But a potential obstacle is that waveforms can be masked due to spatial mixing of several rhythms. We illustrate that neighboring rhythms, as extracted by spatial filtering, can have different waveform properties that are intermixed at the sensor level (see Fig 6). It would be informative to relate these to measures from the microscopic scale, e.g., the firing properties of individual neurons in recordings that have both field recordings and single unit spiking data available [65]. For further analysis of peak frequencies, time domain analysis can help to disentangle harmonic from non-harmonic peaks, e.g., a differentiation between non-sinusoidal properties of the sensorimotor mu-rhythm and genuine beta-bursts, a difference that can be obscured by looking at band-pass filtered signals.

### Physiological considerations: Variability of rhythms across participants

In terms of mapping invasive electrophysiological rhythms, the outlined procedure for investigating dominant rhythms in intracranial data focused on the following methodological considerations: first, electrode activity is always a mixture of many different types of rhythmic and non-rhythmic activity. Therefore, separating putative sources will increase SNR and make it easier to investigate spectral as well as temporal signatures, with e.g., the improved detection of spectral peak frequencies. Not separating sources can result in a "low degree of regional specificity" [5] given that, with a division of channels strictly based on location, volume conduction can lead to a spread of rhythms across regions. Second, the usage of spectral parametrization additionally improves methodological validity in analysis of oscillatory activity through separation from potentially confounding aperiodic activity. Without separation of oscillatory and aperiodic signal contributions, comparing SNR of oscillations in different frequency bands, e.g., for neighboring theta- and alpha band rhythms would require for instance signal whitening, the outcome of which depends on the frequency range used for normalization. By requiring a minimum height of a spectral peak exceeding the aperiodic 1/f-contribution, we ensure to capture oscillatory dynamics.

Equipped with these considerations, we observed that there is high variability in measurable rhythms for individual participants, with for instance no or only weak rhythms in the canonical alpha-frequency range across the sensorimotor cortex, as also observed by [5]. The large degree of variability puts the spotlight on the common practice of electrode pooling, or

combining all electrodes from all participants for analysis, which might inflate false positives due to the contribution of a high number of significant electrodes from a single participant. In light of the observed variability as well as a variable number of electrodes present for each participant, hierarchical models and bootstrapping approaches [66] should be considered in the analysis of intracranial data, to reduce the risk of only a small fraction of participants showing the effects in the group average.

## Limitations

As a general limitation, the estimation of a backward model will never achieve perfect accuracy because dozens of electrodes are not enough to capture the thousands of underlying sources of neuronal activity. One approach for addressing this would be through incorporating simulated iEEG data, where the ground truth is known, such as the LFPy toolbox [67]. Specific limitations of an approach for estimating spatial filters utilizing eigenvalue decomposition are detailed below.

First, there is no automatic one-to-one mapping from estimated components onto physiological entities (but neither can this be done from electrode-based activity). In terms of clarifying what these components represent, [68] have proposed a distinction between genuine, equivalent and representative sources. Within this framework, the components returned by SSD can be seen as representative sources, not directly reflecting e.g. synaptic activity as in the case of genuine sources, but rather presenting one possibility of many, similar to source estimates returned by independent component analysis. In the case of distinct, but highly co-fluctuating neuronal sources, they will not necessarily be separable on the basis of their covariance. An indication of this are spatial patterns that deviate from the spatial pattern expected for a dipolar source, e.g., by showing several spatially distributed maxima. Approaches based on statistical properties of the data (including principal component analysis and independent component analysis) will return as many components as there are electrodes, but not all components will be physiology meaningful in the sense of representing a neuronal source in the frequency band of interest.

In deciding how many components to keep for analysis, the following aspects should be considered when using SSD: Inspecting the relative SNR with aid of the power spectrum is crucial and is simplified because the components are ordered according to SNR in the frequency band of interest. Components without a spectral peak in the frequency band of interest should not be considered when talking about neural oscillations in that specific frequency band [69]. The spatial patterns should be inspected for determining the local focus of the generating source. In addition, bootstrapping approaches based on surrogate data have been suggested to estimate the number of components to retain [44]. Regarding the accuracy of reconstruction for instance in the example in Fig 2, it can be seen that the spatial focus lies on the edge of the recording electrode grid. In that way, the quality of reconstruction is limited by not having more electrodes bordering the spatial maxima. In general, optimizing for SNR and then checking for the presence of high SNR bears the risk of circular analysis. Here, we use a moderate SNR-threshold to retain components. Because our focus on these results is to highlight the degree of variability in the individual recordings, we did not perform a bootstrapping analysis. However, if we wanted to quantify whether the component structure contains more oscillatory structure than expected when running the analysis using spatially correlated 1/f-activity contributions not containing oscillatory bursts, bootstrapping would be appropriate. Circular analysis is not of concern when relative contrasts within conditions are computed, for instance across trials for one participant, where the SSD spatial filters were estimated on the whole data segment, because in this case it is

not the absolute oscillatory power that is crucial, but rather consistent power in- or decreases across experimental conditions.

Further, the estimated spatial filters are invariant with respect to signal polarity, i.e., the sign cannot be uniquely determined. Therefore depending on the choice of parameters, the spatial filter can result in a polarity-inverted signal. For instance, for the participant in Fig 2, the time series and spatial pattern of the second and third component was manually multiplied with -1 for visualization. Alignment of spatially filtered signals can for instance be accomplished according to the sign of the electrode signals, and is straightforward in the case of radially orientated components. In the case of tangentially orientated components, with negative and positive contributions to activity recorded on electrodes, alignment can be made by incorporating knowledge about physiology. Features derived from physiology can include waveform shape, as in the case of the arc-shaped mu-rhythm, or polarity of evoked responses.

Finally, the underlying assumption here is a linear model, and the estimated spatial filters are not dependent on time. This assumption might insufficiently capture traveling wave phenomena, for instance. Propagating activity with high velocity will impact very sharp waveforms, as for electrodes linearly combined with a slight offset a sharp trough will result in a less sharp trough for the component due to time-independent linear combination. For instance, the waveforms in Fig 2 will display a higher peak-trough asymmetry when calculated on high-SNR segments directly from the electrodes, while SSD component traces will have a slightly lower asymmetry measure due to the spatial filtered signal being a linear combination of slightly time-shifted oscillation. It would be of interest for future directions to take wave propagation into account when estimating neuronal oscillatory sources [70, 71].

## Conclusion

Invasive electrophysiological recordings allow for high spatial and temporal resolution investigations into the functional role of the diversity of neural oscillations that are present across the cortex. Different types of oscillations can be seen, showing specific spatial distributions, peak frequencies, waveform shapes and functional modulation, all of which indicate diverse underlying physiology. The spatial and temporal overlap of these rhythms makes the measurement of these different features difficult when only using data derived from single electrodes. Here, we argue that the richness of the data can be better explored when applying data-driven spatial filters, which use multichannel information to specifically enhance the signal-to-noise ratio of oscillations, and therefore improve our ability to study them. This, in turn, helps bridge scales between invasive intracranial measurements and noninvasive, macroscale scalp measurements.

## Supporting information

**S1 Fig. Example: Four different types of spatial filters.** Shown are respectively for rows, bipolar posterior-anterior referencing, bipolar medial-lateral referencing, common average referencing and SSD data-driven referencing: A) Power spectral densities for three channels with the highest SNR in the highlighted band. The gray bar indicates the frequency band defined as the signal contribution for estimating the SSD spatial filters. The power spectrum shows a spectral peak, with additional harmonic peaks. B) The corresponding signal in the time domain showing oscillatory bursts in the alpha-band, amplitudes are normalized for comparison of time courses. For common-average referencing, the red box marks a time period in which less pronounced oscillations can be seen in the common average reference signals, but the oscillatory power of the constituent SSD components is not decreased. C) Electrode grid showing the origin of the electrode signals, for bipolar signals both reference

electrodes are shown, for the common average reference row the center electrodes are shown. For SSD spatial patterns and spatial filters see Fig 2.
(TIF)

**S2 Fig. SSD spatial filters preserve task-related dynamics.** A) Hand-movement trials for the SSD component with highest SNR, with detected burst in the mu-rhythm frequency range highlighted. The event-related desynchronization in the post-cue period is high for hand movements, no desynchronization can be seen for tongue movements, as seen in B). C) The same dynamic can be seen for common average referenced signal for hand movement trials, and tongue movment trials in D). The same burst detection criteria were applied for both referencing methods.
(TIF)

## Author Contributions

**Conceptualization:** Natalie Schaworonkow, Bradley Voytek.

**Data curation:** Natalie Schaworonkow.

**Formal analysis:** Natalie Schaworonkow.

**Funding acquisition:** Bradley Voytek.

**Investigation:** Natalie Schaworonkow, Bradley Voytek.

**Methodology:** Natalie Schaworonkow.

**Project administration:** Bradley Voytek.

**Resources:** Bradley Voytek.

**Software:** Natalie Schaworonkow.

**Supervision:** Bradley Voytek.

**Validation:** Natalie Schaworonkow.

**Visualization:** Natalie Schaworonkow.

**Writing – original draft:** Natalie Schaworonkow, Bradley Voytek.

**Writing – review & editing:** Natalie Schaworonkow, Bradley Voytek.

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
