## [Decision Letter · Decision Letter 0]

26 Apr 2021

Dear Dr Schaworonkow,

Thank you very much for submitting your manuscript "Enhancing oscillations in intracranial electrophysiological recordings with data-driven spatial filters" for consideration at PLOS Computational Biology.

As with all papers reviewed by the journal, your manuscript was reviewed by members of the editorial board and by several independent reviewers.

The paper was overall very well received, still there are some important issues to address and to clarify.

Please make sure that the code can be run by anyone.

In light of the reviews (below this email), we would like to invite the resubmission of a significantly-revised version that takes into account the reviewers' comments.

We cannot make any decision about publication until we have seen the revised manuscript and your response to the reviewers' comments. Your revised manuscript is also likely to be sent to reviewers for further evaluation.

Sincerely,

Daniele Marinazzo

Deputy Editor

PLOS Computational Biology

Daniele Marinazzo

Deputy Editor

PLOS Computational Biology

Reviewer's Responses to Questions

**Comments to the Authors:**

Reviewer #1: I have attached a Word document titled 'Paper Comments' that provides my detailed comments on the paper.

Reviewer #2: In their manuscript entitled "Enhancing oscillations in intracranial elecrophysiological recordings with data-driven spatial filters" the authors Schaworonkow and Voytek report a useful new method and demonstrate its application to a number of datasets. The paper is clearly written, well justified, and the results support the position that this represents a useful tool. I thank the authors for using publically available data and making their code available. I have a few comments in order below that might improve the paper, and believe that it is well suited for publication.

I cloned the repo and but was unable to get the code to run, so I will not comment on it specifically, with the following error.

python3 proc_1_calculate_spectral_param_electrodes.py

hh fixation_pwrlaw

Opening raw data file ../working/hh_fixation_pwrlaw_raw.fif...

Range : 0 ... 130039 = 0.000 ... 130.039 secs

Ready.

Reading 0 ... 130039 = 0.000 ... 130.039 secs...

Effective window size : 3.000 (s)

/Users/kylemathewson/ieeg-spatial-filters-ssd/venv/lib/python3.9/site-packages/matplotlib/text.py:1215: FutureWarning: elementwise comparison failed; returning scalar instead, but in the future will perform elementwise comparison

if s != self._text:

Traceback (most recent call last):

File "/Users/kylemathewson/ieeg-spatial-filters-ssd/code/proc_1_calculate_spectral_param_electrodes.py", line 55, in <module>

fg.plot(

File "/Users/kylemathewson/ieeg-spatial-filters-ssd/venv/lib/python3.9/site-packages/fooof/objs/fit.py", line 622, in plot

plot_fm(self, plot_peaks, plot_aperiodic, plt_log, add_legend,

File "/Users/kylemathewson/ieeg-spatial-filters-ssd/venv/lib/python3.9/site-packages/fooof/core/modutils.py", line 180, in wrapped_func

func(*args, **kwargs)

File "/Users/kylemathewson/ieeg-spatial-filters-ssd/venv/lib/python3.9/site-packages/fooof/plts/fm.py", line 97, in plot_fm

_add_peaks(fm, plot_peaks, plt_log, ax=ax, peak_kwargs=peak_kwargs)

File "/Users/kylemathewson/ieeg-spatial-filters-ssd/venv/lib/python3.9/site-packages/fooof/plts/fm.py", line 147, in _add_peaks

ADD_PEAK_FUNCS[cur_approach](fm, plt_log, ax, **plot_kwargs)

File "/Users/kylemathewson/ieeg-spatial-filters-ssd/venv/lib/python3.9/site-packages/fooof/plts/fm.py", line 265, in _add_peaks_line

freq_point = np.log10(peak[0]) if plt_log else peak[0]

IndexError: invalid index to scalar variable.

Introduction

1) MAJOR. I was a bit confused about the distinction between electrocoritcography, intracranial recordings, and iEEG, and I think later sEEG. I think this paper is focused on electrocorticography (on the surface of cortex), as opposed to depth electrode arrays (intracranial depth electrodes). I think it would help the reader to clarify this early on and in the discussion perhaps indicate how these methods might apply to linear arrays of depth electrodes.

2) MAJOR. Independence of components. I was hoping for some discussion of the relative orthogonality and independence of the components obtained by this method as opposed to simple eigenvalue decomposition, ICA or PCA. No mention or rotation for orthogonality, etc. Are the components obtained independent of one another. The text is often written as if they are (multiple oscillatory sources from same coritical area). Can the authors show mathematically or experimentally how independent their components are and if this is desired.

3) I found most of the math intuitive except the difference between spatial filters and patterns. It seems strange they are different at first glance. I think this is just because the filters are the linear combo if signals needed to derive a source from the data itself. Perhaps a bit more prose regarding your intuitions for why these are different and that is expected would help?

4) Remove artifacts without artifacts from temporal bandpass filtering. I find this hard to buy, seems to good to be true. Maybe this needs to be watered down a bit. Of course one source of "artifact" or a-perfection in the filtering is the width of the band used in the covariance matrix computation (attentuation just outside the filter band). I might argue the imperfection of the filter used to construct the narrow band signal matrix also would introduce analagous artifacts. Perhaps these statements can be tempered a bit.

Methods:

1) bottom of page six, indicate the covariance matrix are over channels (says it later, but useful here)

2) pg. 7 "while the spatial filters are estimated with the aid of covarianc ematrices obtained from narrowband activity, ..." this confused me on first read since they were also made with the braodband noise activity just as much, probably rephrase.

3) OUT THERE. One connection my brain made is that accentuating the size of effects in this way might introduce a dangerous potential to find "voodoo correlations". In that old Vul work, they showed that "using a strategy that computes separate correlations for individual voxels, and reports means of just the subset of voxels exceeding chosen thresholds. We show how this non-independent analysis grossly inflates correlations". Is there a danger of the field over estimating effect sizes if we accentuate the size of the effects in the way proposed here, and what steps should researchers take to avoid this pitfall?

4) Researcher degrees of freedom - I tend to avoid any component selecting in my analysis pipeline, as Laszlo showed that ICA component selection success varries with experiementer experience. Can you imagine more automated ways of selecting components than the heurestics proposed in the paper?

Results:

1) Pg. 15, ln 404, typo "am"

2) Figure 6 - I think e1 and e2 should. be above the comp 1 and 2 to match other figures

3) Figure 7 - can you show a panel here using a normal bandpass noise filter that many people would use, to show the bleed over into neighbouring bands

4) Waveform shape section - I like this and you did show that it was working well after the transformation, but if the hypothesis of this paper is "this technique works better than others" and this section is "it also works for bycycle analysis", then you should probably compare the results to those obtained from non-transformed data.

Discussion:

1) limitations - This section was good but was a bit detached from YOUR analysis and results. Can you point to a couple of your results in each of these limitations. When you say backwards model doesn't "acheive perfect accuracy" - what does this look like in the data, bleed over into components, not separating them well, etc. For the travelling wave example, could this explain any of your results in a different way (multiple sources of alpha from one location)

I think for Figure 7 adding a panel D with a narrow band 60 Hz filter will show how much better yours is.</module>

**Have the authors made all data and (if applicable) computational code underlying the findings in their manuscript fully available?**

Reviewer #1: Yes

Reviewer #2: Yes

PLOS authors have the option to publish the peer review history of their article (what does this mean?). If published, this will include your full peer review and any attached files.

Reviewer #1: **Yes: **Anirudh Wodeyar

Reviewer #2: **Yes: **Kyle E Mathewson
---

## [Decision Letter · Decision Letter 1]

22 Jul 2021

Dear Dr Schaworonkow,

We are pleased to inform you that your manuscript 'Enhancing oscillations in intracranial electrophysiological recordings with data-driven spatial filters' has been provisionally accepted for publication in PLOS Computational Biology.

Best regards,

Daniele Marinazzo

Deputy Editor

PLOS Computational Biology

Reviewer's Responses to Questions

**Comments to the Authors:**

Reviewer #1: I thank the authors for their careful consideration of my comments. I appreciate the thorough responses and the updated manuscript helped me clarify my doubts about the manuscript - particularly with respect to the expectations made about neural sources and also about how to interpret the components relative to these expectations. Further, I think the updated Introduction does in fact help guide the reader along the process of thinking about spatial filters and referencing better.

Reviewer #2: The authors have adequately addressed all of my comments and concerns and I continue to think this is an excellent piece of work well suited for publication, I thank the authors for all the hard work.

**Have the authors made all data and (if applicable) computational code underlying the findings in their manuscript fully available?**

Reviewer #1: Yes

Reviewer #2: Yes

PLOS authors have the option to publish the peer review history of their article (what does this mean?). If published, this will include your full peer review and any attached files.

Reviewer #1: **Yes: **Anirudh Wodeyar

Reviewer #2: **Yes: **Kyle Elliott Mathewson

---

## [Editor Report · Acceptance letter]

16 Aug 2021

PCOMPBIOL-D-21-00438R1 

Enhancing oscillations in intracranial electrophysiological recordings with data-driven spatial filters

Dear Dr Schaworonkow,

I am pleased to inform you that your manuscript has been formally accepted for publication in PLOS Computational Biology. Your manuscript is now with our production department and you will be notified of the publication date in due course.

With kind regards,

Olena Szabo
